# Irreversible inhibitors of the 3C protease of Coxsackie virus through templated assembly of protein-binding fragments

Daniel Becker[1], Zuzanna Kaczmarska[2,3], Christoph Arkona[1], Robert Schulz[1], Carolin Tauber[1], Gerhard Wolber[1], Rolf Hilgenfeld[4,5], Miquel Coll[2,3] & Jörg Rademann[1]

Small-molecule fragments binding to biomacromolecules can be starting points for the development of drugs, but are often difficult to detect due to low affinities. Here we present a strategy that identifies protein-binding fragments through their potential to induce the target-guided formation of covalently bound, irreversible enzyme inhibitors. A protein-binding nucleophile reacts reversibly with a bis-electrophilic warhead, thereby positioning the second electrophile in close proximity of the active site of a viral protease, resulting in the covalent de-activation of the enzyme. The concept is implemented for Coxsackie virus B3 3C protease, a pharmacological target against enteroviral infections. Using an aldehyde-epoxide as bis-electrophile, active fragment combinations are validated through measuring the protein inactivation rate and by detecting covalent protein modification in mass spectrometry. The structure of one enzyme–inhibitor complex is determined by X-ray crystallography. The presented warhead activation assay provides potent non-peptidic, broad-spectrum inhibitors of enteroviral proteases.

[1] Institute of Pharmacy, Medicinal Chemistry, Freie Universität Berlin, Königin-Luise-Straße 2 + 4, 14195 Berlin, Germany. [2] Institute for Research in Biomedicine, Parc Científic de Barcelona, Baldiri Reixac 10-12, 08028 Barcelona, Spain. [3] Institut de Biologia Molecular de Barcelona (CSIC), Parc Científic de Barcelona, Baldiri Reixac 10-12, 08028 Barcelona, Spain. [4] Institute of Biochemistry, University of Lübeck, Ratzeburger Allee 160, 23562 Lübeck, Germany. [5] German Center for Infection Research (DZIF), Hamburg–Lübeck–Borstel Site, University of Lübeck, 23562 Lübeck, Germany. Correspondence and requests for materials should be addressed to J.R. (email: joerg.rademann@fu-berlin.de).

At present, bioactive protein ligands are discovered mostly by the screening of chemical libraries employing a suitable bioassay as a test system. In contrast to classical screening, the templated assembly of protein ligands during the bioassay could be a tantalizing alternative having the potential to identify non-canonical protein ligands with improved potency, ligand efficiency and selectivity, while employing reduced human, ecological and financial resources[1–6]. Protein-templated reactions are driven by the (kinetic) enhancement of the ligand-forming reaction or by the thermodynamic stabilization of the protein-binding ligation product, or by a combination of both. In any of the cases, however, the enhanced binding of a fragment ligation product will be accompanied by a shift of the ligation equilibrium and by the templated and enhanced formation of the bound ligation product. Enhanced binding of the fragment ligation product corresponds to over-additivity of fragment binding, which can be detected sensitively in biochemical and biophysical experiments.

Both reversible[2,7–11] and irreversible[12–20] reactions have been employed for the formation of protein ligands through covalent templated assembly. In this contribution, we investigate the combination of a reversible templated reaction and an irreversible templated reaction, which will enable the protein-guided assembly of protein ligands and their subsequent reaction as irreversible enzyme inhibitors. The described reaction sequence enables the testing of combinations of nucleophilic fragments and electrophilic warheads in a biochemical assay and inhibitory fragment combinations detected in this 'warhead activation assay' are used as blueprints for the construction of potent irreversible enzyme inhibitors. The described warhead activation assay is rationalized by structural and kinetic analysis of the formed fragment ligation products and is supported by molecular modelling of the putative intermediary fragment–protein complexes. Site-specific binding of several inhibitors is verified by protein mass spectrometry and the structure of one enzyme–inhibitor complex is determined by protein crystallography. The most potent inhibitors possess broadband sub-micromolar activity towards a panel of seven entero- and rhinoviral 3C proteases indicating the feasibility of the fragment ligation approach described.

## Results

**Concept of the fragment-warhead activation assay.** The envisaged fragment-warhead activation assay is based on the idea that a templated ligation reaction of a protein-binding fragment with an electrophilic warhead should lead to the accelerated irreversible inhibition of the protein, if a reactive protein nucleophile is present (Fig. 1). In this concept, a nucleophilic small-molecule fragment binds to a binding site of a protein (Fig. 1a, step 1). The protein-binding fragment undergoes a reversible ligation reaction with a bis-electrophilic protein-reactive group (warhead)[21,22] (step 2). The ligation reaction thereby positions the second reactive electrophilic functionality of the warhead at the active site of the protein inducing an irreversible, covalent reaction of the active-site nucleophile with the warhead (step 3) resulting in the irreversible deactivation of the protein. The described reaction sequence enables the testing of combinations of nucleophilic fragments and electrophilic warheads in a biochemical assay and inhibitory fragment combinations detected in this 'warhead activation assay' are blueprints for the construction of potent irreversible enzyme inhibitors.

**Selection of the protein target.** As a protein target, the 3C protease of Coxsackie virus B3 (CVB3 3C protease) was selected[23,24]. Coxsackie viruses are members of the large genus of

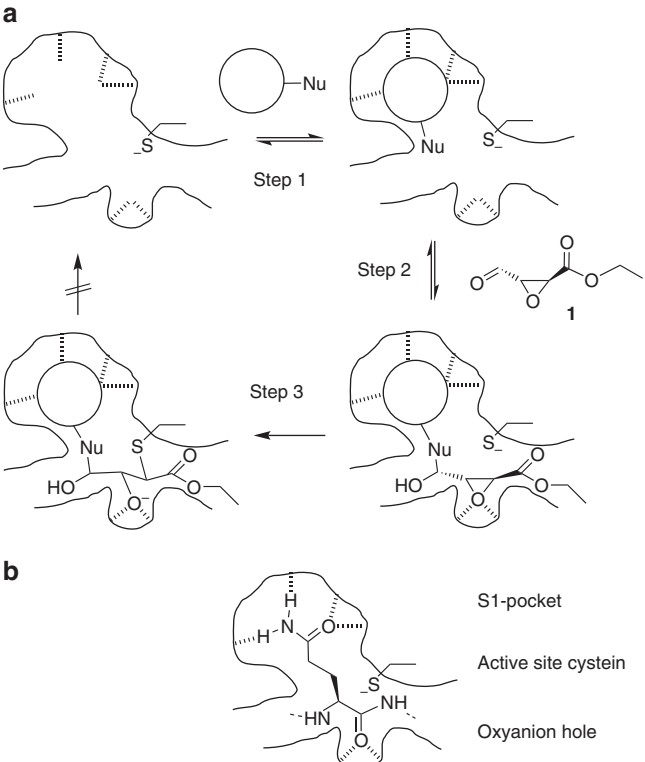

**Figure 1 | Concept of the fragment-warhead activation assay.** (**a**) Protein-templated, covalent assembly of an irreversible inhibitor of a cysteine protease: reversible binding of a nucleophilic fragment (Nu = nucleophile) to the S1-site (step 1) positions the bis-electrophilic warhead **1** in a templated dynamic reaction at the active site of the protein (step 2), leading to irreversible inhibition of the protein (step 3). In the case that Nu is an amine nucleophile (Nu = NHR), an imine intermediate instead of the depicted hemiaminal may be formed. (**b**) The active site of an enterovirus 3C protease with a bound peptide substrate containing a glutamine residue (Gln, Q) in P1 position occupying the S1 pocket of the enzyme.

enteroviruses, representing uncoated positive-strand RNA viruses of the picornavirus family and causing numerous human diseases including meningitis, myocarditis and diarrhea[25–28]. The genus *Enterovirus* comprises numerous human pathogens, including rhinoviruses, polioviruses, echoviruses and other human-infecting viruses like EV-A71 and EV-B93 (refs 26,29,30). Coxsackie virus 3C protease, like other enteroviral proteases, is a cysteine protease with a characteristic specificity for glutamine in the P1 position of its substrates (Fig. 1b). Thus, for this target, fragments binding to the S1 pocket are biomimetics of the glutamine side chain, while the active-site nucleophile is the thiolate of cysteine 147 (Cys-147).

**Development of warhead activation assay.** Several bis-electrophiles were considered as warheads for the warhead activation assay. Enantiomerically pure ethyl (2S,3S)-epoxy-4-oxo-butanoate **1** was selected and prepared from commercially available ethyl 4-oxo-but-(2E)-enoate[31]. This warhead contains both an aldehyde and an epoxide functionality as electrophilic reaction sites for the ligation with a protein-binding, nucleophilic fragment and the irreversible alkylation reaction with the active-site cysteine of the protease. The 2,3-trans-epoxysuccinyl structure is derived from the natural cysteine protease inhibitor E-64 and has been recognized as a privileged reactive warhead targeting cysteine proteases[32–34]. First, the warhead fragment **1**

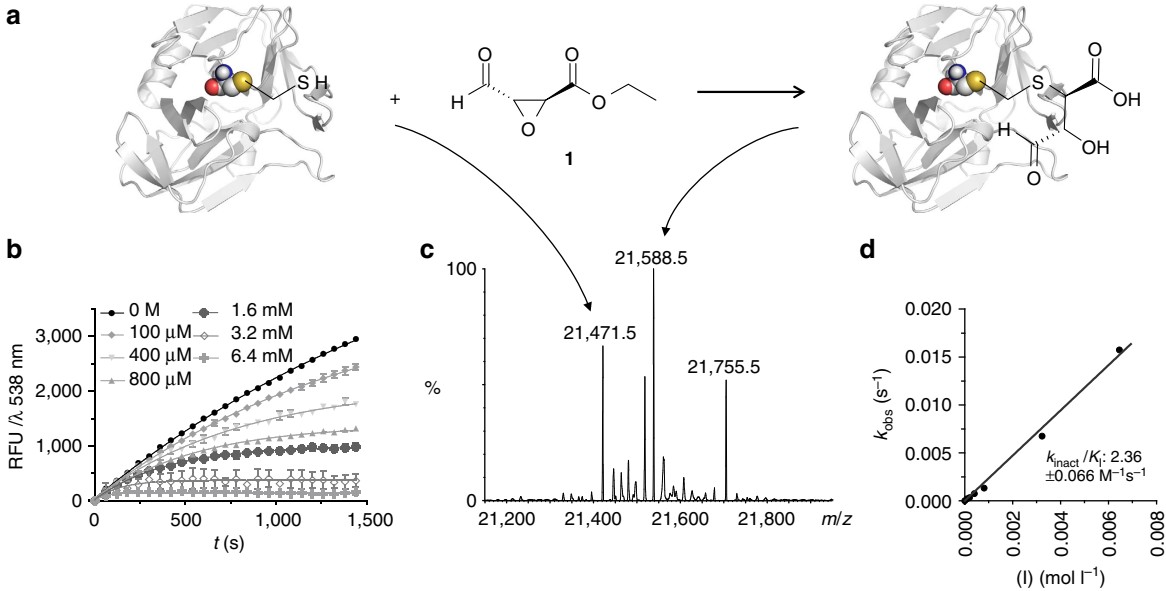

**Figure 2 | Development of the bis-electrophilic warhead 1.** (**a**) Deactivation of Coxsackie virus 3C protease by reaction with bis-electrophilic warhead **1**. Reaction scheme. (**b**) Inhibition of the protease in the FRET-based enzyme activity assay at various concentrations of **1**. (**c**) Deconvoluted ESI-mass spectrum of the reaction of the protease with **1**. The mass of the unmodified protein, 21,471.5 Da, is shifted to a main peak at 21,588.5 Da, corresponding to the mass of the protein $+$**1** after cleavage of the ethyl ester. The peak at 21,755.5 corresponds to the protein mass $+$**1** $+$ Tris (buffer)–water. (**d**) Deactivation of the protease is quantified as relative inhibition, the observed deactivation rate, $k_{obs}$, plotted against inhibitor concentration. The slope of the linear regression curve of the $k_{obs}$ values plotted against the corresponding concentration value yields the $k_{inact}/K_I$-value for compound **1** and equals $2.4 \pm 0.1\,M^{-1}s^{-1}$.

alone was investigated as an inhibitor of the 3C protease of Coxsackie virus B3 (Fig. 2).

Employing an enzymatic protease assay with a substrate containing a fluorescence-resonance energy transfer (FRET) pair, fragment **1** was identified as a weak, irreversible inhibitor of the CVB3 3C protease with a relative inactivation rate ($k_{inact}/K_I$-value) of $2.4 \pm 0.1\,M^{-1}s^{-1}$. This inactivation rate corresponded to an apparent half-maximal inhibitory concentration (IC$_{50}$) value of $\gg 1$ mM after 10 min (Fig. 2b, Table 1). The covalent mode of action was confirmed by electrospray ionization high-resolution ion mobility spectrometry-mass spectrometry (ESI-MS) measurements. Incubation of the protease with warhead **1** shifted the main deconvoluted mass peak of the protein from 21,471.5 to 21,588.5 Da and the observed protein mass difference of 117 Da corresponded well with the molecular weight of bis-electrophile **1** after hydrolysis of the ethyl ester.

Next, a fragment ligation assay was conducted to identify nucleophilic fragments inducing the deactivation of the protein through a ligation reaction with fragment **1**. For this purpose, a concentration of fragment **1** was determined, which decreased the activity of the protease by $\sim 10\%$ over a time period of 10 min. This level of inhibition was used to ascertain the maximal sensitivity of the assay. A small library of 850 primary amine fragments, composed by applying a substructure analysis of the World Drug Index[35], was screened in the fragment ligation assay. Fragment **2**, 5-amino-1-cycloheptyl-1,2-dihydropyrazol-3-one (Fig. 3a, Supplementary Table 1), displayed the strongest over-additive effect of inhibition of protease activity. While fragments **1** or **2** alone reduced the enzyme's activity by no more than 10%, the protease was inhibited completely by a combination of **1** and **2** at the same concentrations (Fig. 3b). Incubation of the protease with mixtures of **1** (1.2 equiv.) and **2** (1 equiv.) led to a concentration-dependent deactivation of the enzyme with an apparent IC$_{50}$ value of ca. 200 μM and a $k_{inact}/K_I$ of $8.3 \pm 0.6\,M^{-1}s^{-1}$ (Supplementary Fig. 1). Deconvolution of the ESI-MS of the protease after ligation with a mixture of both

**Table 1 | Synthesized inhibitors 1–21 and their $k_{inact}/K_I$ or IC$_{50}$ values with Coxsackie 3B virus 3C protease.**

| Compound number | $k_{inact}/K_I\,(M^{-1}s^{-1})$ | IC$_{50}$ (μM) |
|---|---|---|
| 1 | $2.4 \pm 0.1$ | — |
| 2 | 0 | — |
| 3 | $3.7 \pm 0.2$ | — |
| 4 | 0 | $142.2 \pm 5.6$ |
| 5 | 0 | — |
| 6 | 0 | — |
| 7 | $0.42 \pm 0.01$ | — |
| 8 | $131.7 \pm 4.2$ | — |
| 9 | 0 | — |
| 10 | $14.7 \pm 0.9$ | — |
| 11 | 0 | $20.2 \pm 1.6$ |
| 12 | $606.4 \pm 24.9$ | — |
| 13 | $34.7 \pm 1.5$ | — |
| 14 | $63.1 \pm 3.1$ | — |
| 15 | 0 | — |
| 16 | 0 | — |
| 17 | $216.7 \pm 30.7$ | — |
| 18 | $11.7 \pm 0.5$ | — |
| 19 | 0 | — |
| 20 | $1,007.7 \pm 102.2$ | — |
| 21 | $541.5 \pm 6.3$ | — |

fragments **1** and **2** showed the modification of the protein mass by addition of both fragments ($144 + 195 = 339$ Da) as the major mass peak (100% intensity, see Fig. 3c). For comparison, the mass peak for the remaining unmodified protein displayed only 12% intensity and that of the protein coupled to the bis-electrophile **1** alone ca. 8%, supporting the assumption that the protein modification was strongly amplified by the presence of fragment **2**. Incubation of the mutated protease C147A (ref. 36) did not display any modification in MS, indicating a selective alkylation of the active site.

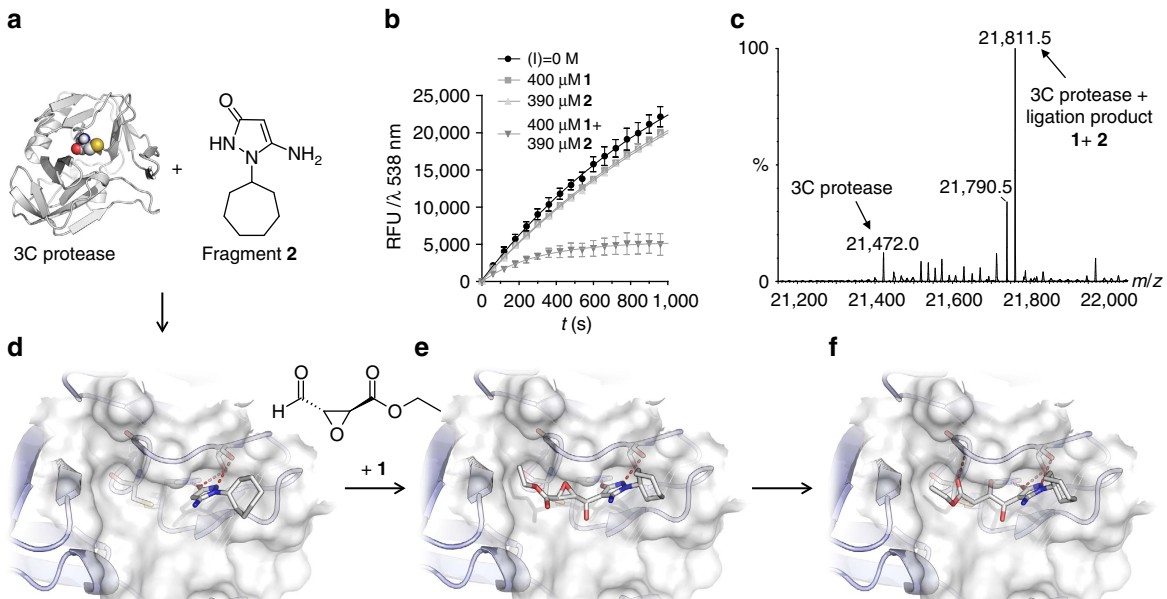

**Figure 3 | Results of screening a fragment library using the warhead activation assay.** (**a**) Molecular modelling of the reversible binding of fragment **2** to the S1 pocket of Coxsackie virus protease, the reversible ligation of warhead **1**, and the irreversible modification of Cys-147. The most important interactions are shown as red dotted lines. (**b**) Over-additive inhibition of Coxsackie virus protease by combination of warhead **1** and fragment **2**. **1** and **2** alone are virtually inactive. (**c**) Deconvoluted ESI-mass spectrum of the 3C protease after incubation with warhead **1** and fragment **2**. Main mass signal at 21,811.5 Da corresponds to the mass of the protein complex $+ 1 + 2$. (**d–f**) Three-dimensional illustrations of fragment binding to the S1 pocket of CV3B 3C protease obtained by docking experiments. (**d**) Initial non-covalent binding of fragment. (**e**) Non-covalent binding of the fragment ligation product. (**f**) Covalent binding of ligation product after epoxide opening.

The ligation reaction of fragments **1** and **2** was analysed by HPLC/MS with and without the protein target present (Supplementary Fig. 2). Several different addition products of **1** and **2** were formed according to mass analysis and retention times and two of these products had the additive mass of both fragments, which had been observed in the deconvoluted MS of the protein sample reacted with both fragments. This formation of several different addition products of **1** and **2** is in agreement with the reported reactivity of 5-amino-1,2-dihydropyrazol-3-ones, which were found to react with electrophiles at both the exocyclic amino group, the carbonyl oxygen and in the carbon-4 position[37,38]. The product ratio varied strongly depending on the solvent and presence/absence of the protein target. Isolation of the formed ligation products by preparative column chromatography furnished again mixtures of compounds, indicating that they were formed reversibly in an equilibrium reaction. To evaluate the potential of the various ligation products as inhibitors, possible fragment combinations were generated *in silico* and docked[39] into the active site. Docking poses were selected according to their ability to bind and react in a similar manner as observed in PDB entry 3ZZA representing the crystal structure of CV3B 3C protease. In particular, two criteria were applied: (i) the hydrogen-bonding pattern between the lactam moiety of the co-crystallized ligand, and His-161 and Thr-142 in the S1 pocket had to be fulfilled by the docked poses for a plausible position in the binding site and (ii) the distance of cysteine sulfur and the potentially covalently binding electrophile of the ligation product had to be in a reasonable range for subsequent covalent binding (Supplementary Table 2). Docking of fragment **2** suggested that **2** can bind to the S1 pocket of the protease occupying the position of the primary amide of the native glutamine residue (Fig. 3d). Sampling the various reaction paths of the electrophile **1** with the nucleophilic positions of fragment **2** supported the best fit for a nucleophilic attack by the carbon-4 ring position (Fig. 3e). According to our docking

experiments, the putative reaction product, a secondary alcohol, can be positioned in the active site of the protein and can be attacked by Cys-147 resulting in the opening of the epoxide ring and the formation of the covalent protein modification (Fig. 3f).

**Fragment-based inhibitor design.** Based on the observation that fragment **2** is a potent activator of bis-electrophile **1** inducing the assembly of an irreversible protease inhibitor, a set of fragment combinations was designed, varying the chemical nature and the attachment of the protein-reactive electrophile (Fig. 4, Table 1, Supplementary Table 1). Trans-epoxide **3** was a moderately active irreversible inhibitor of the protease ($k_{inact}/K_I = 3.7\,\mathrm{M^{-1}\,s^{-1}}$), while trans-epoxide **4** was a reversible inhibitor of the enzyme. Cis-epoxide **5** was inactive as expected, so was the naphtyl-1-methyl amide **6**, which was designed according to a recent report[40]. Increased reactivity with the protease was observed, when alternative electrophiles were linked to the primary fragment **2**. While the acrylic ester derivative **7** was only a weak irreversible inhibitor with $k_{inact}/K_I$ to $0.4\,\mathrm{M^{-1}\,s^{-1}}$, the ethyl fumaroyl electrophile in **8**, containing also a *trans*-olefine, raised the relative inactivation rate to $132\,\mathrm{M^{-1}\,s^{-1}}$, more than 60-fold than the initial electrophile **1**. The same electrophile, an *N*-fumaroyl ester, was inactive in the 5-amino position as shown by compound **9**. Converting the stereochemistry at the double bond of **9** furnished the *N*-maleoyl ester **12** displaying an impressive inactivation rate of $606\,\mathrm{M^{-1}\,s^{-1}}$. To assure that the inhibition of **12** was effected by the amino-pyrazolone fragment, the unsubstituted maleoyl amide ester **13** was prepared and found to be only moderately active ($34\,\mathrm{M^{-1}\,s^{-1}}$).

Transformation of the ethyl ester **12** to the ethyl amide **14** reduced the inactivation rate by 10-fold ($63\,\mathrm{M^{-1}\,s^{-1}}$). The maleoyl-diamide **15** was fully inactive. Further enhancement of the inactivation was accomplished by *trans*-vinyl sulfone **20** displaying the maximal inactivation rate of $1{,}008\,\mathrm{M^{-1}\,s^{-1}}$.

**Figure 4 | Fragment combinations 3–21 prepared as derivatives of the bis-electrophilic warhead 1 and the nucleophilic fragment hit 2.**

Replacement of the 1-*N* substituent cyclohexyl in the most active inhibitors **8**, **12** and **20** by hydrogen, *N*-isopropyl, *N*-phenyl or *N*-hept-4-yl yielded the weaker inhibitors **18**, **21**, **10** and **17**, respectively.

**Broad-spectrum inhibitors of viral 3C proteases**. Three of the most potent inhibitors of CBV 3C protease derived of the S1-binding amino-pyrazolone fragment **2**, maleoyl ester **12**, maleoyl-diamide **14** and vinyl sulfone **20** were tested towards a panel of six additional rhinoviral and enteroviral 3C proteases, equine rhinitis B virus (ERBV-1), human rhinovirus A49 (HRV-A49), human enterovirus D68 (EV-D68), Aichi virus 1 (AiV-1), porcine sapelovirus 1 (PSV-1) and human enterovirus B93 (EV-B93) at protein inhibitor stoichiometries of 1:1 and 1:3 (Fig. 5). Inhibitors **12** and **20** under these conditions strongly reduced the activity of all six proteases investigated, with **20** being generally more potent than **12**. All compounds, **12**, **14** and **20**, showed the highest inhibition for EV-B93 and both **14** and **20** fully inhibited the enzyme even at one inhibitor equivalent. For **12** and **14,** the inhibitory activity decreased for EV-B93 > PSV-1 > ERBV-1 > AiV-1 > EV-D68 > HRV-A49.

IC$_{50}$ values of the most active compounds were determined by incubation of a fixed enzyme concentration for 3 h with different concentrations of inhibitors **14** and **20** (Table 2) and the most potent inhibitor, compound **20**, showed sub-micromolar inhibition for all tested viral proteases, with strongest inhibition for EV-B93, followed by PSV-1, AiV-1, ERBV-1, HRV-A49, and finally

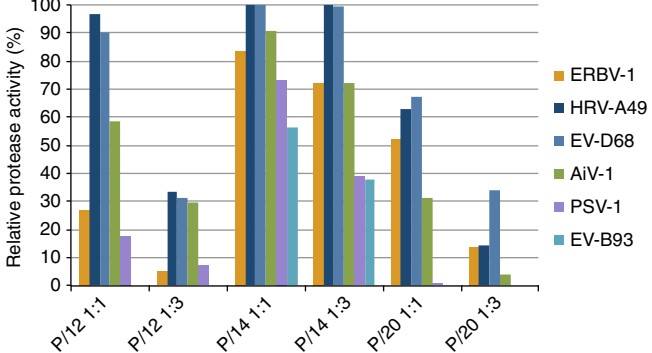

**Figure 5 | Inhibition of 6 different 3C proteases by compounds 12, 14 and 20.** 3C proteases from equine rhinitis B virus (ERBV-1), human rhinovirus A49 (HRV-A49), human enterovirus D68 (EV-D68), Aichi virus 1 (AiV-1), porcine savelovirus 1 (PSV-1) and human enterovirus B93 (EV-B93) were tested at different protease to inhibitor ratios.

HEV-D68, with ERBV-1 being the only protease tested where **12** is more active than **20**.

Finally, the selectivity of the representative inhibitors **14** and **20** was validated using four proteases possessing nucleophilic amino-acid residue in the active site, namely the serine proteases trypsin, chymotrypsin, and factor Xa and the cysteine protease caspase-3. The four enzymes cover a broad range of S1-preferences, namely

**Table 2 | IC$_{50}$ values of inhibitors 14 and 20 towards seven viral 3C proteases.**

| Inhibitor | ERBV-1 | HRV-A49 | HEV-D68 | AiV-1 | PSV-1 | EV-B93 | CBV |
|---|---|---|---|---|---|---|---|
| **14** | >10 μM | ≫20 μM | ≫20 μM | 9 μM | 8 μM | 4 μM | 13 μM |
| **20** | 0.5 μM | 0.3 μM | 1 μM | 0.5 μM | 0.3 μM | 0.2 μM | 0.4 μM |

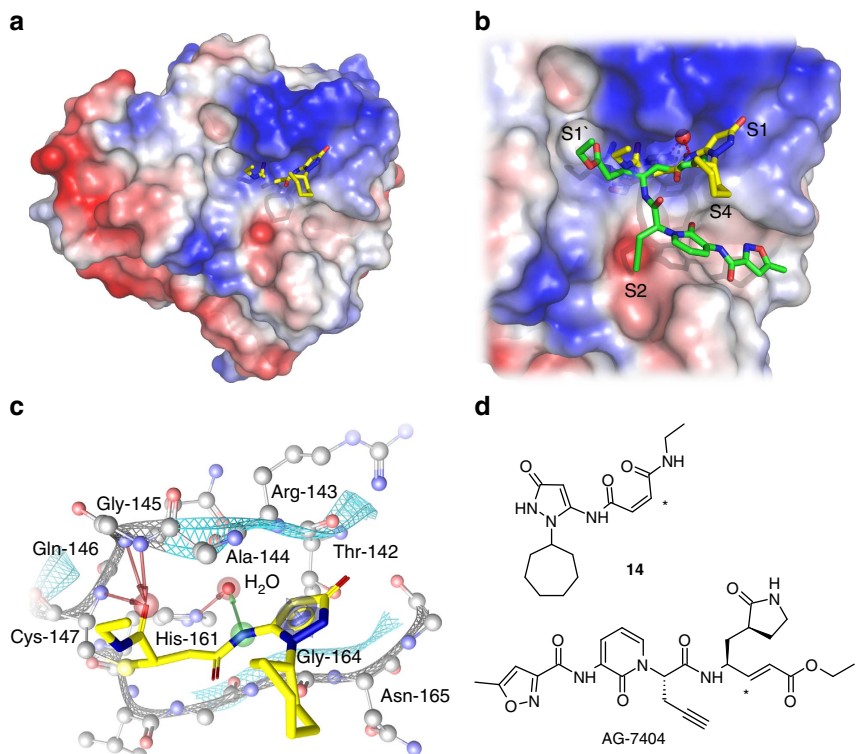

**Figure 6 | Crystal structure of inhibitor 14 with 3C protease of EV-B93 (PDB: 5IYT).** (**a**) Refined structure of the enzyme–inhibitor complex with **14** represented as sticks, carbon atoms in yellow. (**b**) Close-up of the substrate-binding site with bound **14** and AG-7404 (carbon atoms in green); subpockets S1′, S1, S2 and S4 are annotated. Water molecules are represented as red spheres. (**c**) Binding modes of **14**; EV-B93 3C amino-acid residues in close proximity to **14** are labelled in the S1 and S1′ sub-pockets. The arrows represent the H-bonds between **14**, water molecules crucial for inhibitor binding and the protein. (**d**) Structures of inhibitors **14** and AG-7404 with the carbon covalently attacked by the sulfur nucleophile of the protease being highlighted by an asterisk.

for arginine/lysine (trypsin, factor Xa), hydrophobic residues (chymotrypsin) and aspartate (caspase-3). Compound **14** was inactive towards all four proteases (IC$_{50}$ > 500 μM); compound **20** was inactive towards the three serine proteases (IC$_{50}$ > 500 μM) and revealed very low inhibition of the cysteine protease caspase-3 (IC$_{50}$ > 300 μM) (Supplementary Table 3).

**Structure of EV-B93 3C protease with inhibitor 14.** Compounds **12**, **14** and **20** were incubated with EV-B93 3C protease. The enzyme was repurified by size-exclusion chromatography and subjected to crystallization trials. The protein complex with **14** gave well-diffracting crystals, which showed the presence of additional electron density in the active site corresponding to the molecular structure of the inhibitor. The high-resolution structure (PDB: 5IYT, 1.73 Å) of the protease–inhibitor complex revealed several additional aspects of the inhibition of the protease with compound **14**.

As described by our docking studies, exclusively the electrophilic C-2 of **14** was attacked and covalently linked to the nucleophilic Cys-147, forming a stable tetrahedral Michael addition product (Fig. 6a–c). As a result, **14** is inserted between the solvent-exposed β-strand E2 and the loop formed by Thr-142,

Arg-143, and Ala-144 and occupies the S1 and S1′ sub-pockets of the protein. Other than the reference inhibitor AG-7404 (Fig. 6d), which is attacked at C-3 (see asterisk), the C-1 carbonyl oxygen of the ethyl amide in **14** is oriented towards the oxyanion hole and forms a hydrogen-bonding network with the backbone amides of Cys-147, Gln-146 and Gly-145. The N-ethyl group points into the S1′-pocket. The pyrazolone ring of **14** is sandwiched in the S1 pocket by π − π interactions with the peptide bonds between Arg-143—Ala-144 on one side and the Gly-164—Asn-165 on the other side. The pyrazol-5-amide NH interacts with Arg-143, Gly-164, and a water molecule that mediates a hydrogen bond to His-161.

In addition, plausible hydrogen bonds were identified between the pyrazolone ring, and both Gly-164 and Thr-142 as well as to a water molecule that supports the interaction with Ala-144. The cycloheptane shows only weak electron density indicating the flexible orientation of this part of the molecule at the protein surface, a van der Waals interaction is observed between the ring C4 atom and the side-chain C$^β$ atom of Ala-144. In summary, the crystal structure shows that the binding of compound **14** is distinct from that of the established inhibitors such as AG-7404 in several aspects that contribute to an understanding of the observed inhibitory activities. The modified warhead structure

 

seems to favour binding through an additional interaction with the oxyanion hole and with the π-stacking of the pyrazolone. For comparison compounds **12** and **20** were docked to the enzyme and similar binding poses as in the crystal structure were obtained (Supplementary Fig. 3).

## Discussion

In this contribution, a method for the covalent, protein-templated assembly of irreversible enzyme inhibitors is described and implemented. The approach combines a reversible, dynamic fragment ligation reaction with an irreversible covalent modification and inhibition of the target enzyme, Coxsackie virus B3 3C protease. Starting point for the discovery of protein-binding fragments was the bis-electrophilic warhead, epoxy-aldehyde **1**, which combines an electrophilic motif for irreversible inhibition of cysteine proteases, the epoxide, with an aldehyde electrophile for the dynamic ligation of nucleophilic fragments. Warhead **1** alone was only a weak inhibitor of CVB3 3C protease but enabled the sensitive detection of a protein-binding fragment **2**, a pyrazolone, from a collection of nucleophilic fragments. Using a FRET-based assay for protease activity, it could be demonstrated that the reversibly formed ligation product of **1** and **2** led to a 3.5-fold enhancement of the protein inactivation rate from 2.4 to $8.3\,M^{-1}\,s^{-1}$ resulting in the strongly over-additive inhibition of the protease from 10 to 100% inhibition. The templating effect of the protease was confirmed by mass spectrometry showing that inhibition of the protease activity corresponded with shifting the equilibrium to the protein–fragment-warhead complex and thus resulted in the covalent modification of the enzyme's active site.

To develop more potent irreversible inhibitors of enteroviral proteases, the validated hit fragment **2** obtained from the inhibitor assembly assay was modified varying the electrophilic warhead and its attachment to the S1-binding fragment. In this endeavor, starting from an irreversible inhibitor with a relative inactivation rate of $2.4\,M^{-1}\,s^{-1}$ for warhead **1**, the combination with S1-binding fragment **2** furnished potent irreversible inhibitors for several Michael acceptors, most pronounced for (E)-olefines like compound **8** ($k_{inact}/K_I = 131.7\,M^{-1}\,s^{-1}$), (Z)-olefines like compounds **12** and **14** ($k_{inact}/K_I = 606.4$ and $63.1\,M^{-1}\,s^{-1}$, respectively), and vinyl sulfones such as inhibitor **20** ($k_{inact}/K_I = 1,007\,M^{-1}\,s^{-1}$).

Testing of the activity of the inhibitors **12**, **14** and **20** against a panel of six enteroviral proteases indicated that the most potent inhibitors were broad-spectrum inhibitors of this family of genetically related proteases; the most potent inhibitor **20** showed half-maximal inhibitory activities at sub-micromolar concentrations. Broad-spectrum inhibition of enteroviral proteases is highly desirable for the development of drugs that should be applicable in the case of numerous diseases caused by already known and newly emerging enteroviruses.

Compounds **14** and **20** were also validated as inhibitors against a set of four human proteases possessing a nucleophilic active-site residue and were found inactive at concentrations of 100 μM. These results indicate the selectivity of the inhibitors for enteroviral proteases and confirm the selective recognition of the pyrazolone fragment in **14** and **20** by the S1-binding pocket of enteroviral proteases, which selectively bind substrates containing glutamine in P1. On the contrary, the tested serine/cysteine proteases with S1-preference for arginine, as trypsin and factor Xa, or for hydrophobic residues, as chymotrypsin, or for aspartate, as caspase-3, did not bind inhibitors **14** and **20**.

A crystal structure of the complex of EV-B93 3C protease with compound **14** (PDB: 5IYT) finally confirmed the initial hypothesis that protein-binding fragments can be detected in a site-specific manner via a templated ligation reaction with a bis-electrophilic probe to furnish irreversible protease inhibitors.

Beyond the discovery of a class of irreversible, non-peptidic inhibitors of enteroviral 3C proteases, this work has demonstrated a potentially broader concept, how to identify reversibly formed, bioactive fragment ligation products through coupling a dynamic ligation reaction with an irreversible alkylation of the protein target. In this concept, the protein template binding the initial fragment shifts the ligation equilibrium and thereby induces the protein modification reaction. The method should be principally applicable to other protein surfaces provided that the following three conditions are met: a protein-binding nucleophilic fragment, a bis-electrophile (warhead) that is able to bind to the fragment nucleophile and a nucleophilic protein site that enables the irreversible reaction with the bis-electrophile. Consequently, success of the method with alternative protein targets can be expected, if the bound fragment, the bis-electrophile, and the nucleophilic protein site fit spatially and in terms of their chemical reactivity. It should also be noted, however, as demonstrated in the presented work that initially detected fragment-warhead combinations can be optimized further by chemical modification.

## Methods

Protocols and reaction schemes for the chemical synthesis of compounds **1–21** as well as other methodological information are reported in the Supplementary Methods. For characterization and purity of the compounds in this article, see Supplementary Figs 4–29. Identity and purity (>95%) of all compounds were determined by chromatography (silica or RP-18 HPLC), by fully assigned [1]H- and [13]C-NMR spectra and by high-resolution mass spectra. Protein assays were performed with recombinantly expressed viral proteases. Negative control sets (omission of protease), positive control sets (omission of inhibitor), and controls for autofluorescence of substrate or inhibitor were carried out and did not interfere with the assay readout unless noted otherwise. $K_M$- and $v_{max}$-values were determined by double reciprocal plotting of the measured activity against the concentration of substrate (Lineweaver–Burk). Enzyme kinetics were determined with a Fluorimeter SAFIRE II from Tecan (Crailsheim, Germany). Assays were conducted on polystyrene 384-microtiter plates from Corning (model 3,544).

**Mass spectrometry of protein-inhibitor samples.** High-resolution mass spectra were recorded on a Synapt G2-S HDMS by Waters Co. (Milford, UK) equipped with an Acquity UPLC and a Phenomenex Jupiter C-18 column (3 × 250 mm, 5 μm, 300 Å). The ionization and transfer parameters of the MS system have been optimized for a standard sample (insulin) and were applied without modification. Deconvolution of measurement results was performed by the MaxEnt1 method of the MassLynx software by Waters Co. The most important source parameters are: capillary voltage: 3.3 kV, sampling cone: 40 V, source offset: 60 V, source temperature: 100 °C, desolvation temperature: 250 °C, desolvation gas flow: 600 l h[−1]. Chromatographic conditions: start 95% solvent A, within 35 min linear gradient to 1% solvent A, 5 min 1% solvent A, within 4 min back to 95% solvent A. Solvent composition: A—water, 0.1% formic acid; B—acetonitrile, 0.1% formic acid. Flow rate: 0.5 ml min[−1]. For mass spectra of protease–inhibitor complexes in this article, see Supplementary Figs 30–35.

**Recombinant protein production and purification.** The enzymatically active 3C protease of Coxsackie virus B3 consisting of 183 amino-acid residues and carrying an additional C-terminal sixfold histidine tag was expressed from the plasmid pET-23a(+) (Novagen). The cDNA of the 3C protease corresponds to GenBank accession no. M88483.1. Protein production and purification followed standard conditions. Briefly, a clone of BL21(DE3) E. coli cells transfected with the coding plasmid was grown in LB medium in the presence of ampicillin (100 μg ml[−1]) until $OD_{600nm}$ reached 0.6. Expression was induced by adding IPTG to a final concentration of 1 mM and further incubation was carried out for 4 h at 37 °C. Cells were harvested by centrifugation and lysed by ultrasonication in 25 mM Tris-HCl pH 7.5/50 mM NaCl/1% (v/v) Triton X-100/10% (v/v) glycerol/2 mM DTT. The extract was clarified by centrifugation and separated on NiNTA agarose (Qiagen) in 50 mM Tris-HCl pH 8.0/500 mM NaCl applying a step gradient of 50 to 500 mM imidazole. Eluates of highest volume activity were pooled and further separated on a preparative Superdex 75 column (GE Healthcare) equilibrated in 25 mM HEPES-NaOH pH 7.5/50 mM NaCl. The final pool of recombinant protease was adjusted to 20% (v/v) glycerol and frozen in aliquots at −80 °C until use.

EV-B93 3C protease expression and purification was conducted as previously described[41]. Other proteases were obtained from the European Virus Archive (http://www.european-virus-archive.com/).

**Table 3 | Peptide sequences used for testing inhibitory activity of 3C proteases.**

| Protease | Peptide sequence |
|---|---|
| ERBV-1 3C | Ac-AKDGGIFAQ↓SRDRHYLVGTV-NH$_2$ |
| HRV-A49 3C | Ac-RHSVGATLEALFQ↓GPPVYREIKIS-NH$_2$ |
| EV-D68 3C | Ac-RHSVGATLEALFQ↓GPPVYREIKIS-NH$_2$ |
| AiV-1 3C | Ac-RHSVGATLEALFQ↓GPPVYREIKIS-NH$_2$ |
| PSV-1 3C | Ac-RHSVGATLEALFQ↓GPPVYREIKIS-NH$_2$ |
| EV-B93 3C | Ac-RHSVGATLEALFQ↓GPPVYREIKIS-NH$_2$ |

**Table 4 | Data collection and refinement statistics for the complex of EV-B93 3C protease with inhibitor 14 (PDB: 5IYT).**

| | EV-B93 3C protease-14 |
|---|---|
| *Data collection* | |
| Space group | P2$_1$ |
| Cell dimensions | |
| a,b,c (Å) | 57.46, 52.33, 62.27 |
| α, β, γ (°) | 90.00, 103.73, 90.00 |
| Resolution (Å) | 50.00–1.73 (1.77–1.73)* |
| $R_{merge}$ | 0.056 (0.743) |
| I/σI | 13.05 (2.14) |
| Completeness (%) | 97.1 (96.5) |
| Redundancy | 2.7 (2.7) |
| | |
| *Refinement* | |
| Resolution (Å) | 30.00–1.73 |
| No. of reflections | 34,781 |
| $R_{work}$/$R_{free}$ | 0.187/0.217 |
| No. of atoms | |
| Protein | 2,873 |
| Ligand | 23 |
| Water | 182 |
| B-factors | |
| Protein | 28.73 |
| Ligand | 61.36 |
| Water | 40.18 |
| R.m.s.d.'s | |
| Bond lengths (Å) | 0.006 |
| Bond angles (°) | 1.398 |

*Values in the parentheses are for the highest resolution shell. A single crystal was used for data collection.

**Assay for template-assisted assembly of protease inhibitors.** The activity of the protease was measured by hydrolysis of the FRET-substrate *N*-Dabcyl-KTLEALFQ↓GPPVYE(Edans)-NH$_2$. The peptide was purchased from Biosyntan (Berlin, Germany). Samples were irradiated with light with a wavelength of 355 nm and the emitted light at 538 nm was measured in relative fluorescence units (RFUs). An amount of 100 pmol of cleaved substrate equals 5,000 RFU.

Assay samples contained 10 µl of a 100 mM HEPES-NaOH buffer solution pH 7.5/1 mM EDTA, 10 µl enzyme solution (2 µM CVB3 3C$^{Pro}$), 10 µl of 10 µM substrate solution (both dissolved in the above buffer) and 1 µl Inhibitor and/or fragment dissolved in DMSO. For assay results of a dilution series of a mixture of **1** and **2** and comparison of reaction products in dependence of the assay solvent, see Supplementary Figs 1 and 2.

The screening procedure consisted of the following steps: 10 µl 100 mM HEPES-NaOH buffer solution pH 7.5/1 mM EDTA was placed in a cavity of the 384-well plate (polystyrene, Corning 3544). Then 10 µl of enzyme solution (40 µM CVB3 3C protease in 100 mM HEPES-NaOH buffer solution, pH 7.5/1 mM EDTA) was added and the plate was shaken for 2 min at 1,400 r.p.m. Afterwards, 0.5 µl of a DMSO solution of the electrophilic probe and 0.5 µl of a DMSO solution of the nucleophile was added, followed by 10 µl of substrate solution (10 µM in 100 mM HEPES-NaOH buffer solution, pH 7.5/1 mM EDTA). The plate was shaken again for 2 min at 1,400 r.p.m. and the generated fluorescence signal was measured with a Safire II Fluorimeter from Tecan. The electrophilic probe **1** was used at a concentration of 400 µM for the screening, which lowered the protease activity by 10–20%. Eight-hundred fifty primary amines were tested at a concentration of 320 µM and fragment **2** amplified the inhibitory activity of **1** most potently. Positive control samples contained only DMSO instead of the probe or the nucleophile. Negative control samples contained no active enzyme. An interfering fluorescent effect has not been observed.

When determining the inhibitory parameters of an inhibitor, the assay set-up was as mentioned but 1 µl DMSO solution of inhibitor was used instead of probe or nucleophile. Different concentrations of inhibitor were tested at least as duplicates and the pseudo-first-order association kinetics ($k_{obs}$) were determined using GraphPad Prism. The slope of a plot of the different $k_{obs}$ values against the inhibitor concentration yielded $k_{inact}/K_I$.

***In vitro* proteolytic activity assay.** A volume of 100 µl of reaction mixture containing 1.9 µM 3C protease, 25 µM substrate peptide, 5% DMSO, buffer (30 mM Tris pH 8.0, and 50 mM NaCl or 50 mM HEPES pH 7.4 and 50 mM NaCl) and (optionally) different concentrations of inhibitor was incubated in 37 °C during 3 h. The reaction was quenched by addition of 0.5% (final concentration) TFA. Samples were analysed by HPLC on SOURCE 15RPC ST 4.6/100 column using: 2–90% linear gradient of ACN in 0.1% TFA. The 3C protease activity in the absence of inhibitor was defined as 100%. The following peptides (GL Biochem, Shanghai, China) were dissolved in DMSO at a concentration of 5 mM and used for testing 3C proteases activity (Table 3).

**Reaction of EV-B93 3C protease with inhibitor 14.** The EV-B93 3C protease was mixed with a 12-fold molar excess of the inhibitor and incubated overnight at 30 °C. The complex was purified by gel filtration (Superdex 75 10/300, GE Healthcare) in 10 mM Tris, pH 8.0, 200 mM NaCl, 2 mM DTT. The collected fractions were concentrated to 2.3 mg ml$^{-1}$ for crystallization.

**Crystallization and structure determination.** Crystals were grown by the sitting-drop vapour diffusion method at 20 °C by mixing equal volumes of protein solution and reservoir solution containing 25% polyethylene glycol 3,350, 0.2 M ammonium acetate, and 0.1 M Bis-Tris, pH 6.5. Crystals were transferred to cryoprotectant solution comprising mother liquor and 25% glycerol for a few seconds and flash-cooled in liquid nitrogen. X-ray diffraction data were collected at beamline XALOC at the ALBA synchrotron (Cerdanyola del Vallès, Barcelona) (Table 4). Diffraction data were indexed and integrated using XDS[42], and scaled and merged with SCALA[43]. Structure elucidation and refinement were carried out using the CCP4 suite of programs[44]. The initial phases were obtained by molecular replacement using PHASER[45] with the previously solved EV-B93 3C structure as search model (PDB code: 3Q3X)[41]. REFMAC[46] was employed for structure refinement and Coot[47] for interpretation of electron density and model building. Geometry restraint information for **14** was generated with SKETCHER within CCP4. For a stereo image of the electron density map of the binding site of the protease–inhibitor complex see Supplementary Fig. 36. The crystal structure was validated with MolProbity[48] and showed 98.1% of all residues in Ramachandran-favoured and 100.0% in Ramachandran-allowed regions.

**Protein ligand docking analysis.** Protein Data Bank[49] entry 3ZZA was used as structure for the CVB3 3C protease. For a detailed binding confirmation of inhibitors **12** and **20** by ligand docking, see Supplementary Fig. 3. Docking was performed using GOLD v 5.0.2 by the Cambridge Crystallographic Data Center (Cambridge, UK, Europe) after protein preparation using MOE version 2,014.09 (ref. 50). The binding site was defined using a radius of 10 Å around the sulfur atom of cysteine 147. Docking was carried out using the PLP scoring function[51] with 25 GA runs per ligand in the most accurate mode without water molecules and default parameter settings. Covalent docking was performed by constraining the sulfur atom in the ligand and cysteine 147 to share the same position coordinates. Docking poses were analysed and prioritized for S1 pocket binding and putative covalent bond formation using 3D pharmacophore models developed in LigandScout version 4.0 (ref. 52) based on crystal structures of the CVB3 3C protease (3ZZ6, 3ZZ7, 3ZZ8, 3ZZ9, 3ZZA, 3ZZB). Graphical representations of selected conformations were generated using the PyMOL Molecular Graphics System, Version 1.7.4 Schrödinger, LLC.

**Data availability.** Data supporting the findings of this study are available within the article and its Supplementary Information files and from the corresponding author upon reasonable request. The crystal structure presented within this article is deposited under PDB code 5IYT.

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

## Acknowledgements

This work was supported by the European Commission (Cooperation Project SILVER, GA No. 260644), the Deutsche Forschungsgemeinschaft (project B9 within the SFB 765), the *Ministerio de Economia y Competitividad* (Grants BFU2011-22588 and BFU2014-53550-P to M.C.) and the *Generalitat de Catalunya* (Grant 2014-SGR-01530 to M.C.).

## Author contributions

J.R., D.B. and C.A. conceived and designed the experiments, D.B., C.A. and Z.K. performed the experiments, D.B., Z.K., C.A., R.S., M.C., G.W. and J.R. analysed the data. R.H. and C.T. contributed reagents. J.R., D.B., Z.K. and R.S. wrote the manuscript.

## Additional information

**Competing financial interests:** The authors declare no competing financial interests.

**Reprints and permission** information is available online at http://npg.nature.com/ reprintsandpermissions/

