## [Peer Review File · Nature Communications]

Transferred manuscripts:

Reviewers' comments:

Reviewer #1 (Remarks to the Author):

- A, This is a meritorious ms that describes a novel strategy for identifying protein-binding fragments via the combination of a reversible templated reaction and an irreversible templated reaction. The strategy was subsequently validated using Coxsackie virus B3 3C protease as a target.
- B. The findings are timely and original and should be of wide interest and applicability.
- C. The ms is well-written and the methodologies employed are well-presented. The non-peptidic inhibitors identified are weak to fair inhibitors of the enzyme.
- D. Appropriate
- E. The conclusions are valid and clearly-presented and defended.
- F. The ms can be strengthened considerably by evaluating the selectivity of a representative inhibitor using a panel of human proteases that includes members of all five classes of proteases).
- G. Appropriate
- H. This is a very well-written ms that is characterized by clarity, and high significance and innovation.

Reviewer #2 (Remarks to the Author):

This paper describes the development of irreversible inhibitors of the 3C protease using template assembly of protein-binding fragments. This builds on previous research of the Rademann research group who have explored 'kinetic' guided template assembly using a range of proteins. The '3C protease' is an interesting target and there has been previous structural studies carried out by Hilgenfeld, who is a co-author on this paper. The concept of the paper involves the binding of a nucleophilic fragment which undergoes a reaction with a reactive bis-electrophilic 'warhead' and forms a covalent bonds though epoxide ring opening and the with reactive cysteine which is present in the binding pocket.

- While this is an interesting concept there is the question of how applicable this to other enzymes (other than proteases) and this needs to be highlighted?
- The enantiomerically pure epoxide 'warhead' is described however there is no indication as to where and why this structural motif was chosen. Does this have to be enantiomerically pure?
- The small library of 850 nucleophilic fragments were screened in the ligation assay however there is no indication as the composition of this focused library. Were these randomly chosen fragments?
- The examination of the protein after modification using ESI-MS show that there is 'a small fraction of the protein remained unmodified' and 'little of the protein was coupled to the bis electrophile', this needs to be more specific and a percentage rather than the description is needed.
- A small amount of SAR was explored around the bis-electrophilic warhead and nucleophilic fragment hit and two compounds were identified 4 and 11 which have IC50 of 142 and 30 uM respectively which are not strongly binding.
- Compound 20, which showed the highest inactivation for 3C protease also showed potent inhibition across a panel of six other proteases, however there is no IC50 value for the 3C protease, why was this not measured?

Overall this is an interesting concept however the applicability of this to other targets beyond

proteases is difficulty to ascertain. The discovery of the reactive nucleophilic and electrophilic species in such close proximity might be difficult and this would encounter problems typically associated with fragment linking and in-situ click chemistry strategies.

Reviewer #3 (Remarks to the Author):

J Rademann and co-workers report on the identification of protein-binding fragments via protein-templated assembly using the Coxsackie virus B33C protease as a relevant drug target. The initially discovered weak inhibitor of this Cys protease is subsequently optimized into a potent inhibitor through careful design and the binding mode is validated by protein crystallography.

The method presented should in principle be generally applicable to the discovery of covalent inhibitors for any protein target, going beyond the class of Cys proteases. However, so as to be able to use the key concept underlying the strategy, a nucleophilic side chain needs to be present in the binding pocket.

The data and methodology are presented in a very clear manner, meaning that the results obtained are highly valid.

The conclusions are clear and do not make claims that go beyond realistic extensions of the approach. In other words, the limitations, namely the presence of a reactive amino acid side chain in proximity of the binding pocket.

The references are generally speaking fine. Two key references should be added: 1) In addition to reference 4, I would recommend adding a reference to the following paper: A. Herrmann, Chem. Soc. Rev., 2014, 43, 1899.

2) To acknowledge conceptually related approaches, a short section referring to the report by D. A. Erlanson and co-workers (Bioorg. Med. Chem. Lett., 2008, 18, 3978-3981) should be added.

Below a list of suggestions for improvement:

p.3: the second paragraph, in which the target is introduced, should also provide the reader with information on any known inhibitors. This will help to appreciate the results better.

p.4: at pH = 7.5, imine formation can take place in water, so the question is whether the hemiaminal is formed or the imine or perhaps even a mixture of both. If the imine predominates, it is obviously not a good biomimetic of the natural substrate's Gln that is bound in the same region of the active site. This question should be verified experimentally if not done already.

p.5: when introducing fragment 1, the choice of having one reversible (with the bound fragment) and one irreversible reaction (with the enzyme) should be explained. Furthermore, is there not a risk of "inverted reactivity", namely that the nucleophilic fragment reacts with the epoxide and that the Cys side chain forms a hemithioacetal with the aldehyde of fragment 1. Has this issue been investigated?

p.6: the reasoning behind choosing conditions such that protease activity is decreased by 10% by fragment 1 should be explained.

p.6: please provide more detail on the type of library that was used.

p.6-7 repeating this experiment in the presence of a known ligand targeting the active site would further confirm the selective alkylation of the active site.

p.8: Fig. 3 only has sub figures A-C, in the text, the authors refer to sub-figures D-F.

p.8: provide at least some detail in the main text on the docking experiments.

p.8: the expression "molecular simulation" might be misleading as the reader might be thinking of molecular dynamics simulations. The authors presumably refer to the docking study. This section should be rephrased.

p.8: how were the other fragment combinations designed? It would also be important to explain why the authors chose to synthesise the fragment combinations individually rather than performing a second screen in which the protein would template inhibitor formation using new fragments and warheads?

p.12: can the authors explain why only a crystal structure in complex with ligand 14 was obtained and not with ligand 20, which is the more potent inhibitor.

p.12: "As predicted by molecular modeling": do the authors again refer to their docking studies? If so, please rephrase. Given the lack of co-crystal structures of the target in complex with compounds 12 and 20, it would be very informative to repeat the same docking studies with those compounds.

p.15: could the authors comment whether it is positive or negative that the inhibitors identified are broad-spectrum inhibitors? A reader not familiar with the field would otherwise struggle to appreciate this.

Supporting Information:

1. For any compounds subjected to biochemical assays, proof of purity needs to be provided!
2. Report melting points as temperature ranges

And some minor grammatical mistakes and typos.

p.1: correct: "Here, a strategy..."

p.8: correct: "Figure 4, Table 1)"

p.9: correct: "cycloheptyl" (unless there is a mistake in the structures in Figure 4)

p.14: correct: "Warhead 1 alone is only a weak..."

p.15: correct: "potent inhibitors are broad-spectrum..."

SI:

no space before % throughout the SI

use upper-case L for liter throughout the SI

Dear reviewers,

Thank you very much for your detailed, careful, and supportive comments, which we would like to reply to in the following point-by-point.

Reviewer #1 (Remarks to the Author):

A, This is a meritorious ms that describes a novel strategy for identifying protein-binding fragments via the combination of a reversible templated reaction and an irreversible templated reaction. The strategy was subsequently validated using Coxsackie virus B3 3C protease as a target.

B. The findings are timely and original and should be of wide interest and applicability.

C. The ms is well-written and the methodologies employed are well-presented. The non-peptidic inhibitors identified are weak to fair inhibitors of the enzyme.

D. Appropriate

E. The conclusions are valid and clearly-presented and defended.

F. The ms can be strengthened considerably by evaluating the selectivity of a representative inhibitor using a panel of human proteases that includes members of all five classes of proteases).

We have tested the representative inhibitors **14** and **20** with a panel of four non-viral proteases, namely trypsin, factor Xa, chymotrypsin, and caspase-3:

“Finally, the selectivity of the representative inhibitors **14** and **20** was validated using four proteases possessing nucleophilic amino acid residue in the active site, namely the serine proteases trypsin, chymotrypsin, and factor Xa and the cysteine protease caspase-3. The four enzymes cover a broad range of S1-preferences, namely for arginine/lysine (trypsin, factor Xa), hydrophobic residues (chymotrypsin) and aspartate (caspase-3). Compound **14** was inactive with all four proteases ($IC_{50} > 500 \mu\text{M}$); compound **20** was inactive with the three serine proteases ($IC_{50} > 500 \mu\text{M}$) and revealed very low inhibition of the cysteine protease caspase-3 ($IC_{50} > 300 \mu\text{M}$) (Supplementary Table 3).”

G. Appropriate

H. This is a very well-written ms that is characterized by clarity, and high significance and innovation.

Reviewer #2 (Remarks to the Author):

This paper describes the development of irreversible inhibitors of the 3C protease using template assembly of protein-binding fragments. This builds on previous research of the Rademann research group who have explored 'kinetic' guided template assembly using a range of proteins. The '3C protease' is an interesting target and there has been previous structural studies carried out by Hilgenfeld, who is a co-author on this paper. The concept of the paper involves the binding of a nucleophilic fragment which undergoes a reaction with a reactive bis-electrophilic 'warhead' and forms a covalent bonds though epoxide ring opening and the with reactive cysteine which is present in

the binding pocket.

- The enantiomerically pure epoxide 'warhead' is described however there is no indication as to where and why this structural motif was chosen. Does this have to be enantiomerically pure?

The enantiomerically pure epoxide was selected on the basis of the cysteine protease inhibitor E-64, which contains the epoxide motif in the same stereochemistry. For details please see also refs. 30-32.

- The small library of 850 nucleophilic fragments were screened in the ligation assay however there is no indication as to the composition of this focused library. Were these randomly chosen fragments?

The small fragment library was a collection of primary amines composed using a maximum common substructure (MCS) of the world drug index as described in our publication ref 35. In short, the WDI was analyzed for cyclic substructures that occur frequently in bioactive molecules resulting in approx. 570 core fragments and fragment combinations. For the library commercially available amine fragments were selected representing a diverse substructure composition similar to that of the WDI.

- The examination of the protein after modification using ESI-MS show that there is 'a small fraction of the protein remained unmodified' and 'little of the protein was coupled to the bis electrophile', this needs to be more specific and a percentage rather than the description is needed.

Thank you for this advice. We changed the description accordingly.

A small amount of SAR was explored around the bis-electrophilic warhead and nucleophilic fragment hit and two compounds were identified 4 and 11 which have IC₅₀ of 142 and 30 μ M respectively which are not strongly binding.

Yes, we agree.

- Compound 20, which showed the highest inactivation for 3C protease also showed potent inhibition across a panel of six other proteases, however there is no IC₅₀ value for the 3C protease, why was this not measured?

Thanks for this hint. We have added the data requested.

Overall this is an interesting concept however the applicability of this to other targets beyond proteases is difficult to ascertain. The discovery of the reactive nucleophilic and electrophilic species in such close proximity might be difficult and this would encounter problems typically associated with fragment linking and in-situ click chemistry strategies.

The presented concept in principal is applicable to every protein with an active nucleophilic residue positioned in fitting proximity to a fragment-binding pocket and thus is not limited to proteases, hydrolases, and transferases. There are several reports in

the literature that nucleophilic protein residues react specifically with electrophiles not only at active sites but also adjacent to ligand binding pockets, e.g. as demonstrated in the works of Hamachi et al. Nature Chemical Biology 2009 and JACS 2012.

Reviewer #3 (Remarks to the Author):

J Rademann and co-workers report on the identification of protein-binding fragments via protein-templated assembly using the Coxsackie virus B33C protease as a relevant drug target. The initially discovered weak inhibitor of this Cys protease is subsequently optimized into a potent inhibitor through careful design and the binding mode is validated by protein crystallography.

The method presented should in principle be generally applicable to the discovery of covalent inhibitors for any protein target, going beyond the class of Cys proteases. However, so as to be able to use the key concept underlying the strategy, a nucleophilic side chain needs to be present in the binding pocket.

We completely agree. A reactive nucleophile is required in or adjacent to the binding pocket.

The data and methodology are presented in a very clear manner, meaning that the results obtained are highly valid.

The conclusions are clear and do not make claims that go beyond realistic extensions of the approach. In other words, the limitations, namely the presence of a reactive amino acid side chain in proximity of the binding pocket.

The references are generally speaking fine. Two key references should be added: 1) In addition to reference 4, I would recommend adding a reference to the following paper: A. Herrmann, Chem. Soc. Rev., 2014, 43, 1899.

2) To acknowledge conceptually related approaches, a short section referring to the report by D. A. Erlanson and co-workers (Bioorg. Med. Chem. Lett., 2008, 18, 3978-3981) should be added.

The suggested references were added to the manuscript; see ref. 6 and 20.

Below a list of suggestions for improvement:

p.3: the second paragraph, in which the target is introduced, should also provide the reader with information on any known inhibitors. This will help to appreciate the results better.

Has been done.

p.4: at pH = 7.5, imine formation can take place in water, so the question is whether the hemiaminal is formed or the imine or perhaps even a mixture of both. If the imine

predominates, it is obviously not a good biomimetic of the natural substrate's Gln that is bound in the same region of the active site. This question should be verified experimentally if not done already.

In the specific case described in the manuscript, i.e. the ligation of protein with warhead 1 and fragment 2, we found a mass of 339 m/z in the MS corresponding exactly to the mass of the hemiaminal.

We have, however, shown before (ref. 8) that N, S, and O-nucleophiles can undergo templated reactions on proteins, and C-nucleophiles might be reactive, too.

Therefore, we decided to depict a general representation of the concept in Figure 1 using "Nu" as a general abbreviation for a nucleophile. In the legend we have added the note that in case of amines (Nu=NHR) a hemiaminal or an imine can be formed during the reaction.

p.5: when introducing fragment 1, the choice of having one reversible (with the bound fragment) and one irreversible reaction (with the enzyme) should be explained. Furthermore, is there not a risk of "inverted reactivity", namely that the nucleophilic fragment reacts with the epoxide and that the Cys side chain forms a hemithioacetal with the aldehyde of fragment 1. Has this issue been investigated?

Indeed as described by the referee bis-electrophiles in principal can react with two nucleophiles in different ways. In the specific case studied here, however, we have not observed any experimental indication for the postulated "inverted reactivity". There might be several reasons excluding the inverted reaction of the warhead 1 with protein and fragment 2. Firstly, the epoxide in 1 is a much less reactive electrophile than the aldehyde. Secondly, the amino-pyrazolone 2 is as a heteroaromatic amine a much weaker nucleophile than the active-site thiol. Therefore, opening of 1 will occur exclusively or much faster with the thiol nucleophile and not with the aromatic amine or another weaker nucleophile. If reversible reactions of the cysteine thiol with the aldehyde occur, these would not be problematic, as they proceed reversibly in equilibrium leading to now stable irreversibly formed product.

p.6: the reasoning behind choosing conditions such that protease activity is decreased by 10% by fragment 1 should be explained.

We chose conditions with 10% inhibition in order to ascertain maximal sensitivity of the assay and to have the largest possible measurement window. A note explaining this has been added to the manuscript.

p.6: please provide more detail on the type of library that was used.

See the remarks to referee 2:

The small fragment library was a collection of primary amines composed using a maximum common substructure (MCS) of the world drug index as described in our publication ref (35). In short, the WDI was analyzed for cyclic substructures that occur frequently in bioactive molecules resulting in ca. 570 core fragments and fragment combinations. For the library commercially available amine fragments were selected representing a diverse substructure composition similar to that of the WDI.

p.6-7 repeating this experiment in the presence of a known ligand targeting the active site would further confirm the selective alkylation of the active site.

We have used a mutant protease carrying the Cys-Ala mutation in order to prove the selective alkylation of the active site cysteine.

p.8: Fig. 3 only has sub figures A-C, in the text, the authors refer to sub-figures D-F.

Has been corrected.

p.8: provide at least some detail in the main text on the docking experiments.

Details on docking and selection of the docking poses were added to the manuscript.

p.8: the expression "molecular simulation" might be misleading as the reader might be thinking of molecular dynamics simulations. The authors presumably refer to the docking study. This section should be rephrased.

Throughout the text the term "molecular simulation" was replaced by the specific computational methods applied (mostly docking).

p.8: how were the other fragment combinations designed? It would also be important to explain why the authors chose to synthesise the fragment combinations individually rather than performing a second screen in which the protein would template inhibitor formation using new fragments and warheads?

Having identified the S1-binding fragment 2 the fragment combinations were designed in order to define the best attachment point for the warhead at hit fragment 1 and in order to optimize the warhead reactivity with the cysteine residue. As we had already found the S1-binding fragment, for the warhead variation we found the synthesis of fragment combination the simpler and more practical approach. One reason is that we have had no warhead library available and the preparation of such a library would have been another major project.

p.12: can the authors explain why only a crystal structure in complex with ligand 14 was obtained and not with ligand 20, which is the more potent inhibitor.

In principle, every minor change of the protein, e.g. the covalent attachment of another inhibitor, or of any other experimental conditions can have a strong effect on the outcome of the crystallization experiment. Obviously we were able to generate the protein modified by reaction with ligand 20, however, under the studied conditions no crystallization occurred. One possible explanation is that the solubility of 14 in water was better than that of 20.

p.12: "As predicted by molecular modeling": do the authors again refer to their docking studies? If so, please rephrase. Given the lack of co-crystal structures of the target in complex with compounds 12 and 20, it would be very informative to repeat the same

docking studies with those compounds.

Has been rephrased. Docking studies with compounds 12 and 20 have been conducted and added to the SI part (Supplementary Figure 3, Supplementary Table 2).

p.15: could the authors comment whether it is positive or negative that the inhibitors identified are broad-spectrum inhibitors? A reader not familiar with the field would otherwise struggle to appreciate this.

We have added the following sentence to the discussion:
Broad-spectrum inhibition of enteroviral proteases is highly desirable for the development of drugs that should be applicable in the case of numerous diseases caused by already known and newly emerging enteroviruses.

Supporting Information:

1. For any compounds subjected to biochemical assays, proof of purity needs to be provided!

Identity and purity (>95%) of all compounds were determined by chromatography (silica or RP-18 HPLC), by fully assigned ¹H- and ¹³C NMR spectra (see supplementary NMR spectra) and by high-resolution mass spectra. The NMR spectra have been added to the Supplementary Information for every compound. If there was insufficient sample amount for a carbon NMR spectrum or an indistinct proton NMR spectrum, a reversed phase HPLC chromatogram of the purified compound is also provided.

2. Report melting points as temperature ranges
This has been corrected.

And some minor grammatical mistakes and typos.

p.1: correct: "Here, a strategy..."

p.8: correct: "Figure 4, Table 1)"

p.9: correct: "cycloheptyl" (unless there is a mistake in the structures in Figure 4)

p.14: correct: "Warhead 1 alone is only a weak..."

p.15: correct: "potent inhibitors are broad-spectrum..."

All corrected.

SI:

no space before % throughout the SI

use upper-case L for liter throughout the SI

We have deleted the space before %. To our knowledge Nature journal use l for liter.

We hope we were able to answer your questions to your full satisfaction and remain with best regards,

Jörg Rademann

REVIEWERS' COMMENTS:

Reviewer #1 (Remarks to the Author):

The revised ms has been strengthened considerably. The authors have addressed the concerns of the reviewers and have carried out additional experiments to ascertain the selectivity of the inhibitors. Overall, the revised ms is highly meritorious and has attained the standards expected of the journal. Publication is strongly recommended.

Reviewer #4 (Remarks to the Author):

The authors have responded adequately to comments from the reviewers. The paper is technically sound. However, in this reviewer's opinion, the significance of the results is a bit oversold. Bis-electrophile 1, an epoxyaldehyde, is a weak irreversible inhibitor of the target protease. In the presence of nucleophile 2, an amine, the inactivation rate increases four-fold (hardly dramatic or exciting). Then various amides representing a covalent combination of 1 + 2 are made and tested. The really good ones are no longer epoxides, but rather Michael-acceptor olefins. This is tantamount to a bait-and-switch in terms of selling the main idea of the paper, which seems to be that positioning an accessory nucleophile near the active site affords a general way to screen for structural elements (fragments) with which to design good inhibitors.

Point-by-point response to reviewers:

Reviewer #1 (Remarks to the Author):

The revised ms has been strengthened considerably. The authors have addressed the concerns of the reviewers and have carried out additional experiments to ascertain the selectivity of the inhibitors. Overall, the revised ms is highly meritorious and has attained the standards expected of the journal. Publication is strongly recommended.

Reviewer #4 (Remarks to the Author):

The authors have responded adequately to comments from the reviewers. The paper is technically sound. However, in this reviewer's opinion, the significance of the results is a bit oversold.

We have carefully checked the text throughout the manuscript. Wherever possible the precision of our statements has been increased in order to avoid any impression of overselling.

Bis-electrophile **1**, an epoxyaldehyde, is a weak irreversible inhibitor of the target protease. In the presence of nucleophile **2**, an amine, the inactivation rate increases four-fold (hardly dramatic or exciting).

The addition of nucleophile **2** raises the inactivation rate of **1** from $2.4 \text{ M}^{-1}\text{s}^{-1}$ to $8.3 \text{ M}^{-1}\text{s}^{-1}$ corresponding to a 3.5-fold increase. As a result of this increase the inhibition of the protease raises from 10% to 100%. This observed, very significant increase in inhibition enabled us to detect fragment **2** sensitively as an S1-site binding ligand of the protease, which is not detectable in standard inhibition assays. In order to clarify this point and increase the precision, the corresponding statement in the discussion section has been rephrased to:

“Using a FRET-based assay for protease activity, it could be demonstrated that the reversibly formed ligation product of **1** and **2** led to a 3.5-fold enhancement of the protein inactivation rate from 2.4 to $8.3 \text{ M}^{-1}\text{s}^{-1}$ resulting in the strongly over-additive inhibition of the protease from 10 to 100% inhibition.”

Then various amides representing a covalent combination of **1** + **2** are made and tested. The really good ones are no longer epoxides, but rather Michael-acceptor olefins.

Fully agreed. We have carefully described the optimization of the reactive electrophile from epoxides to Michael acceptors in the manuscript. The fact that the potency of our inhibitors increased by modifying the warhead, however, does not discredit our screening strategy but confirms the validity of the fragment hit **2**. In addition, it is rather expectable that changing the warhead would also change the activity of the inhibitors.

This is tantamount to a bait-and-switch in terms of selling the main idea of the paper, which seems to be that positioning an accessory nucleophile near the active site

affords a general way to screen for structural elements (fragments) with which to design good inhibitors.

The main idea of the paper is stated in the abstract clearly related to the enteroviral proteases as targets and has been modified slightly to increase precision:

“Here we present a strategy that identifies protein-binding fragments through their potential to induce the target-guided formation of covalently bound, irreversible enzyme inhibitors. A protein-binding nucleophile reacts reversibly with a bis-electrophilic warhead, thereby positioning the second electrophile in close proximity of the active site nucleophile of a viral protease, resulting in the covalent de-activation of the enzyme. The concept is implemented for Coxsackie virus B3 3C protease, a pharmacological target against enteroviral infections.”

The applicability of the concept to alternative proteins is considered in the discussion part and has been modified in order to describe the limitations and prerequisites for using the approach with other protein targets:

“The method should be principally applicable to other protein surfaces provided that the following three conditions are met: a protein-binding nucleophilic fragment, a bis-electrophile (warhead) that is able to bind to the fragment nucleophile and a nucleophilic protein site that enables the irreversible reaction with the bis-electrophile. Consequently, success of the method with alternative protein targets can be expected, if the bound fragment, the bis-electrophile, and the nucleophilic protein site fit spatially and in terms of their chemical reactivity. It should also be noted, however, as demonstrated in the presented work that initially detected fragment-warhead combinations can be optimized further by chemical modification.”